# Depletion of WWP1 Increases Adrb3 Expression and Lipolysis in White Adipose Tissue of Obese Mice

**DOI:** 10.3390/ijms26094219

**Published:** 2025-04-29

**Authors:** Yuka Nozaki, Yuko Ose, Chinatsu Ohmori, Yuhei Mizunoe, Masaki Kobayashi, Akiyoshi Saitoh, Yoshikazu Higami

**Affiliations:** 1Laboratory of Molecular Pathology and Metabolic Disease, Faculty of Pharmaceutical Sciences, Tokyo University of Science, Chiba 278-8510, Japan; 2Laboratory of Pharmacology, Faculty of Pharmaceutical Sciences, Tokyo University of Science, Chiba 278-8510, Japan; 3Department of Nutrition and Food Science, Graduate School of Humanities and Sciences, Ochanomizu University, Tokyo 112-8610, Japan; 4Institute for Human Life Science, Ochanomizu University, Tokyo 112-8610, Japan; 5Division of Cell Fate Regulation, Research Institute for Biomedical Science, Tokyo University of Science, Chiba 278-1501, Japan

**Keywords:** WWP1, Adrb3, lipolysis, white adipose tissue

## Abstract

Obesity is defined as abnormal or excessive accumulation of body fat and contributes to several metabolic disorders. White adipose tissue (WAT) releases energy as free fatty acids and glycerol from triglycerides through a process called lipolysis. People with obesity have impaired catecholamine-stimulated lipolysis, but comprehensive understanding of this lipolysis is still unclear. We previously showed that expression of WW domain-containing E3 ubiquitin ligase 1 (WWP1), a member of the HECT-type E3 family of ubiquitin ligases, was increased in WAT of obese mice. In this study, we generated *Wwp1* knockout (KO) mice to evaluate the effect of WWP1 in WAT of obese mice. The mRNA levels of beta-3 adrenergic receptor (*Adrb3*), which were decreased with a high-fat diet, were increased by *Wwp1* KO in WAT. Moreover, *Wwp1* KO mice showed increased phosphorylated hormone-sensitive lipase levels in WAT. In contrast, noradrenaline and its metabolism were not altered in WAT of obese *Wwp1* KO mice. These findings indicate that WWP1, which is increased in adipocytes because of obesity, is a candidate for suppressing lipolysis independently of noradrenaline metabolism. We anticipate that inhibition of WWP1 is a promising approach for a new treatment of obesity and type-2 diabetes using Adrb3 agonists.

## 1. Introduction

White adipose tissue (WAT) is the most effective lipid storage organ and plays an important role in systemic metabolic homeostasis. White adipocytes, which are the main component of WAT and contain unilocular lipid droplets, act as an endogenous energy supply by increasing lipolysis during periods of deficient nutritional availability. During the development of obesity, WAT becomes hypertrophied to store excess lipid as triglycerides (TGs) in white adipocytes. WAT in obesity is infiltrated by a large number of macrophages, and this recruitment is associated with metabolic disorders, such as insulin resistance and type-2 diabetes, through chronic inflammatory or oxidative stress [1,2].

Obesity is associated with an increase in basal lipolysis [3,4] but a decrease in catecholamine-stimulated lipolysis [5]. Resistance to catecholamine-induced lipolysis in WAT has been demonstrated in people with obesity [4,6]. These data suggest that decreased expression of the β3-adrenergic receptor, which is encoded by the *Adrb3* gene, and hormone-sensitive lipase (HSL), which in turn causes impaired lipolytic capacity of WAT, leads to metabolic disorders in people with obesity. Mirabegron and vibegron, which have been approved as Adrb3 agonists for treating overactive bladder, have the basic pharmacophore 2-amino-1-phenylethanol, similar to adrenaline [7,8,9,10]. These Adrb3 agonists have been developed by several pharmaceutical companies as treatments for obesity and type-2 diabetes but have yet to become successful.

WW domain-containing E3 ubiquitin ligase 1 (WWP1) (also known as TIUL1 or AIP5) belongs to the homologous to the E6-associated protein carboxyl terminus (HECT)-type E3 ubiquitin protein ligase family. WWP1 has a C2 domain at its N-terminal, four WW domains (for recognition of PPXY-containing substrate) in its central region, and a HECT domain at its C-terminal (for the transfer of ubiquitin to substrate) [11,12]. We previously reported that WWP1 expression levels increase in a p53-dependent manner in a mouse model of obesity caused by a high-fat diet (HFD) [13]. To evaluate the involvement of WWP1 in metabolic regulation, we previously generated and studied systemic *Wwp1* knockout (KO) mice and found that *Wwp1* KO mice had improved insulin sensitivity and glucose tolerance from obesity-related insulin resistance [14]. In addition, we reported that obese *Wwp1* KO mice showed enhanced hepatic insulin signaling and improved hepatic steatosis [15]. These results suggest that WWP1 contributes to systemic glucose metabolic regulation. However, whether WWP1 is involved in lipid homeostasis is unknown. Therefore, in this study, we investigated lipid homeostasis in WAT of obese mice by analyzing systemic KO of *Wwp1*.

## 2. Results

### 2.1. Depletion of WWP1 Increases Adrb3 mRNA Expression in Epididymal WAT

We found that HFD-induced obesity increased WWP1 protein expression in WAT, similar to our previous report [13]. Under HFD-fed conditions, *Wwp1* KO mice exhibited reduced body weight, while there were no difference in subcutaneous WAT (sWAT) or epididymal WAT (eWAT) mass and amount of food intake compared with WT mice (Figure 1A,B and Table 1). The depletion of WWP1 in eWAT of *Wwp1* KO mice was observed at the protein level (Figure 1). To gain insight into the distinctive features of eWAT in *Wwp1* KO mice, we conducted a comparative transcriptome analysis using RNA-seq (Figure 2A). The 33 genes upregulated by *Wwp1* KO with a normal diet (ND) or HFD were nervous-system-related genes, which suggests that depletion WWP1 contributes to nervous-system-related pathways. Among them, 12 genes are related to peripheral tissue (Figure 2B). Pathway enrichment analysis using DAVID showed that the genes involved in nervous-system-related pathways were increased by *Wwp1* KO only in the HFD condition (Figure 2C). Differentially expressed genes (DEGs) were then determined using the edgeR program package [16]. *Wwp1* KO enriched genes related to “cellular response to cAMP” and “energy reserve metabolic process” in HFD mice (Figure 2D). Obesity is frequently associated with chronic inflammation. We found that immune-system-related genes were not altered in *Wwp1* KO (Figure 2D). We also found that an HFD-induced decrease in *Adrb3* expression compared with the ND condition in WT mice was significantly increased in *Wwp1* KO mice (Figure 2A,E). 

### 2.2. Depletion of WWP1 Increases Phosphorylation of HSL in eWAT and Plasma Non-Esterified Fatty Acid (NEFA) Concentrations

We performed Western blotting and analyzed the amount of lipid-homeostasis-related protein expression in *Wwp1* KO mice in the ND or HFD condition. We found that the ratio of phosphorylated HSL (S660)/HSL was increased (suggesting enhanced lipolysis) in *Wwp1* KO mice not only in the ND, but also in the HFD condition (Figure 3). Notably, plasma NEFA concentrations were increased only in HFD-fed *Wwp1* KO mice (Figure 4). However, levels of lipogenesis-related proteins, namely acetyl-CoA carboxylase (ACC), ATP citrate lyase (ACLY), and malic enzyme 1 (ME1), were not significantly different between *Wwp1* KO and WT mice (Appendix A). These results suggested that depletion of WWP1 enhanced lipolysis by increasing phosphorylation of HSL in the HFD-fed model.

### 2.3. Noradrenaline Metabolism Is Not Changed by Wwp1 KO in eWAT

To evaluate whether noradrenaline (NA) contributes to the lipolytic response of *Wwp1* KO mice, we measured the levels of NA and its end metabolite 3-methoxy-4-hydroxyphenilglycol (MHPG) in eWAT derived from ND- and HFD-fed *Wwp1* KO mice. NA and MHPG levels were not significantly different between *Wwp1* KO and WT mice in the ND- and HFD-fed conditions (Figure 5A,B), and the ratio of MHPG/NA was also not different between the two groups of mice in either diet condition (Figure 5C).

## 3. Discussion

β3-adrenergic receptor protein, which is coded by the *Adrb3* gene, is stimulated by NA and then increases intracellular cAMP levels through activation of adenylyl cyclase. cAMP then activates protein kinase A, and protein kinase A catalyzes the phosphorylation of HSL to increase lipolysis [17]. Recent human cohort studies have shown that Adrb3 expression in subcutaneous WAT is negatively correlated with the body mass index in women with or without obesity [6]. In addition, several previous studies have shown that obesity reduces Adrb3 expression [18,19] and intracellular cAMP levels in adipocytes of WAT [18]. We found that *Adrb3* mRNA expression was decreased by an HFD in WT mice, and this HFD-induced downregulation was completely abolished with *Wwp1* KO (Figure 2E). We also found that upregulated genes by depletion of WWP1 enriched the “cellular response to cAMP”-related and “energy reserve metabolic process”-related genes using GSEA (Figure 2D),and increased phosphorylated HSL levels (Figure 3) and plasma NEFA concentrations (Figure 4). On the basis of our previous finding that an HFD increased WWP1 expression [13], we propose that WWP1 is one of the candidates that negatively regulates lipolysis owing to decreased Adrb3 expression in WAT during obesity. However, further studies are required to clearly identify the relationship between WWP1 and Adrb3 expression.

We also evaluated downstream signaling to assess whether there is cAMP-PKA signaling in WAT of obese *Wwp1* KO mice. Although no significant change in global PKA substrate phosphorylation was detected in our study (Appendix A), cAMP/PKA signaling is compartmentalized within subcellular microdomains and coordinated by A-kinase anchoring proteins, which may not be reflected in whole-tissue lysates [20,21]. Therefore, compartmentalization and local control of cAMP may be essential for effective lipolytic signaling downstream of Adrb3, with potential modulation by WWP1.

WWP1 is important for developing neurons and controlling the nervous system. WWP1 and WWP2, which are mediated by SRY-box9, contribute to the polarization of developing pyramidal neurons [22]. In addition, Qin et al. suggested that WWP1 interacts with Nogo-1, which is a key factor of inhibitory central nervous system regeneration and maintaining the integrity of the neuromuscular junction [23]. In our study, nervous-system-related pathways were increased by *Wwp1* KO (Figure 2B,C). Based on these results, we speculated that WWP1 has the potential to control NA metabolism in peripheral nerves in WAT, but we found that NA metabolism did not change (Figure 5). Therefore, enhancing lipolysis in WAT due to WWP1 deficiency is mainly due to signal transduction via increased expression of Adrb3 in adipocytes and might not involve the nerve transmission and NA metabolism in WAT.

In our previous study, we showed that obese *Wwp1* KO mice improved whole-body glucose metabolism compared with WT mice using the insulin tolerance test (ITT) and glucose tolerance test (GTT) [14]. To assess the role of WWP1 in glucose metabolism, in a previous study we evaluated insulin signaling (pAkt/Akt rate) in insulin-sensitive tissues, including WAT, the liver, and skeletal muscle. We found that obese *Wwp1* KO mice showed an enhanced hepatic insulin signaling response and had a reduced weight and triglyceride contents only in the liver and not in WAT [15]. Therefore, targeting WWP1 may be useful in the treatment of obesity and type-2 diabetes through its dual functions. One of these functions is improving systemic glucose metabolism by increasing hepatic insulin sensitivity, and the other is increasing lipolysis in WAT by increasing Adrb3.

## 4. Materials and Methods

### 4.1. Mice

All mouse experiments and protocols were conducted in accordance with the Fundamental Guidelines for Proper Conduct of Animal Experiments and Related Activities in Academic Research Institutions under the jurisdiction of the Ministry of Education, Culture, Sports, Science and Technology of Japan. The study protocol was approved by the Ethics Review Committee for Animal Experimentation at Tokyo University of Science (approval numbers: Y20043, Y21043, and Y22037). Mice with systemic KO of *Wwp1* (*Wwp1*^−/−^ mice) and wild-type (WT) *Wwp1*^+/+^ mice were generated as previously reported [14]. Genotyping of the offspring was performed by polymerase chain reaction (PCR) using KOD FX neo (Toyobo, Osaka, Japan) with the following primers: forward, 5′-AGA GGC AAG AGA ATG GCG TCA AG-3′; and reverse, 5′-GGA GGT GAA AGG GTT GGA AGA ATA C-3′. The mice were maintained under specific-pathogen-free conditions at 23 °C, under a 12 h light/dark cycle in the animal facility at the Faculty of Pharmaceutical Sciences, Tokyo University of Science. They had free access to water and were fed a Charles River Formula-1 diet (21.9% crude protein, 5.4% crude fat, and 2.9% crude fiber; Oriental Yeast, Tokyo, Japan). At 5 weeks old, WT and KO mice were divided into the ND group and the HFD group. The Charles River Formula-1 diet and High-Fat Diet 32 (25.5% crude protein, 32.0% crude lipids, and 2.9% crude fiber; CREA, Tokyo, Japan) were fed as the ND and HFD, respectively. At 15 weeks old, the mice were euthanized under isoflurane anesthesia (Mylan, Canonsburg, PA, USA) and organs were collected. These tissues were immediately diced, frozen in liquid nitrogen, and stored at −80 °C.

### 4.2. RNA-Seq and Gene Set Enrichment Analysis

Total RNA was extracted from eWAT derived from ND-WT (*n* = 4), ND-*Wwp1* KO (*n* = 3), HFD-WT (*n* = 4), and HFD-*Wwp1* KO (*n* = 4) mice using ISOGEN II (Nippon Gene, Toyama, Japan) according to the manufacturer’s protocol. Library preparation and RNA-seq were outsourced to Tsukuba i-Laboratory LLP (Ibaraki, Japan). The RNA-seq reads were aligned to the mouse reference genome GRCm39, which was calculated using STAR two-pass alignment for mapping RNA-seq reads to genomes [24,25]. The number of transcripts per million was calculated and gene expression levels were quantified using featureCounts from the Subread package [26]. DEGs were calculated using the edgeR program package [16].

### 4.3. Immunoblotting

The preparation of cell lysates and immunoblotting were performed according to our previously reported methods [27]. Briefly, cells were lysed with sodium dodecyl sulfate sample buffer, boiled for 5 min, and sonicated. Lysates were subjected to sodium dodecyl sulfate–polyacrylamide gel electrophoresis (15 µg protein/well), and separated proteins were transferred onto nitrocellulose membranes. The membranes were blocked with blocking solution (2.5% skim milk, 0.25% bovine serum albumin in Tris-buffered saline with Tween 20 (25 mM Tris-HCl, pH 7.4, 140 mM NaCl, 2.5 mM KCl, and 0.1% Tween 20)) for 1 h at room temperature and then probed with the appropriate primary antibodies overnight at 4 °C. The anti-WWP1 antibody was generated in our laboratory [13]. Anti-HSL (#4017), anti-phospho-HSL (S660, #4126), ACC (#3662) antibodies, and phospho-PKA substrate (#9624) were from Cell Signaling Technology (Danvers, MA, USA). Anti-ACLY (#1699-1) and anti-phospho-ACLY (#1822-1) antibodies were from Epitomics, an Abcam company (Cambridge, UK), The anti-FASN antibody (fatty acid synthase; #910963) was from BD Biosciences (Franklin Lakes, NJ, USA). The anti-ME1 antibody (#SAB4501853) was from Sigma-Aldrich (St. Louis, MO, USA), and the anti-lamin B1 antibody (#PM064) was from MBL (Aichi, Japan). Subsequently, the membranes were incubated with appropriate secondary antibodies (horseradish-peroxidase-conjugated F[ab′]2 fragment of goat anti-mouse immunoglobulin G or anti-rabbit immunoglobulin G (Jackson ImmunoResearch; West Grove, PA, USA)) for 1 h at room temperature. Antibody-bound proteins were visualized using ImmunoStar LD Reagent from Wako (Osaka, Japan) and a LAS3000 Image Analyzer from Fujifilm (Tokyo, Japan) or ChemiDoc Touch MP from Bio-Rad (Hercules, CA, USA). The data were analyzed using MULTIGAGE software (v3.2) (GE Healthcare, Chicago, WI, USA) or Image Lab (v6.1.0) from Bio-Rad (Hercules).

### 4.4. Image Processing of Original Blots

The original blots are shown in the Appendix A. Original images of full-length membranes could not be included because the membranes were cut before hybridization with antibodies. Therefore, all replicates performed for some blots are included.

### 4.5. Quantitative Real-Time PCR

Total RNA was extracted using ISOGEN II (Nippon Gene), and reverse transcription was performed using ReverTra Ace^®^ qPCR RT Master Mix (Toyobo). Quantitative real-time PCR was performed using the CFX Connect^TM^ Real Time System (Bio-Rad, Hercules) and Thunderbird SYBR qPCR mix (Toyobo) according to the manufacturers’ protocols. Values were normalized to Rps18 expression levels. All expression changes were normalized to the ND-WT. Sequences of primers used for PCR were as follows: *Adrb3* (forward, 5′-AAC TGA AAC AGC AGA CAG GGA C-3′ and reverse, 5′-CCC CCA TGT ACA CCC TAG TT-3′), and *Rps18* (forward, 5′-TGC GAG TAC TCA ACA CCA ACA T-3′ and reverse, 5′-CTT TCC TCA ACA CCA CAT GAG C-3′). *Rps18* was used as a housekeeping gene.

### 4.6. Plasma Non-Esterified Fatty Acid Concentrations

Plasma non-esterified fatty acid (NEFA) concentrations were measured using LabAssay NEFA (Fujifilm Wako Pure Chemical, Osaka, Japan) and this was performed in accordance with the manufacturer’s protocols.

### 4.7. Quantification of Monoamine Contents in the Brain

Using eWAT, MHPG and NA levels were quantified using high-performance liquid chromatography. The eWAT was homogenized in 500 µL of 0.2 M perchloric acid containing 2 ng of isoproterenol as an internal standard. The homogenates were placed on ice for 30 min and then centrifuged at 15,000 rpm for 20 min at 4 °C. To maintain the mixture at pH 3, 0.1 M sodium acetate was added to the supernatant. The samples were filtered using 0.2 µm Minisart RC 4.0 (Sartorius, Göttingen, Germany). The samples (10 µL) were analyzed by high-performance liquid chromatography using electrochemical detection. NA levels were quantified under the following conditions. An electrochemical detector (ECD-300; Eicom Co., Ltd., Kyoto, Japan) was equipped with a graphite electrode (WE-3G; Eicom Co., Ltd.), which was used at a voltage setting of 650 mV with an Ag/AgCl reference electrode. The mobile phase comprised 0.1 M sodium acetate/0.1 M citric acid buffer (pH 3.5) containing 12% methanol, 175 mg/L 1-octanesulfonic acid sodium salt, and 5 mg/L EDTA-2 Na. Monoamines were separated on a C-18 column (150 × 3 mm reversed-phase, EICOMPAK SC-5ODS; Eicom Co., Ltd.). The mobile phase flow rate was maintained at 0.3 mL/min with a column temperature of 30.0 °C. MHPG levels were quantified under the following conditions. The electrochemical detector (ECD-300; Eicom Co., Ltd.) was equipped with a graphite electrode (WE-3G; Eicom Co., Ltd.), which was used at a voltage setting of 750 mV with an Ag/AgCl reference electrode. The mobile phase comprised 0.1 M sodium acetate/0.1 M citric acid buffer (pH 3.5) containing 13% methanol, 180 mg/L 1-octanesulfonic acid sodium salt, and 5 mg/L EDTA-2 Na. Monoamines were separated on a C-18 column (150 × 3 mm reversed-phase, EICOMPAK SC-5ODS; Eicom Co., Ltd.). The mobile phase flow rate was maintained at 0.5 mL/min with a column temperature of 30.0 °C [28].

### 4.8. Statistical Analysis

Statistical analysis was performed using GraphPad Prism version 10.1.2 (324) and version 8.4.1 (676) (GraphPad Software, San Diego, CA, USA; www.graphpad.com). To assess the normality of data distribution in each group, we conducted the Shapiro–Wilk test and visual inspection using Q-Q plots (Appendix A). While the Shapiro–Wilk test did not provide statistically significant evidence of normality, likely because of the small sample size, the Q-Q plots did not indicate substantial deviations from the normal distribution. Therefore, we judged that the data were approximately normally distributed. An ANOVA is robust for mild violations of normality, particularly in small-sample studies [29,30]. Therefore, a two-way ANOVA was applied to compare groups.

## 5. Conclusions

Obesity is associated with an increase in basal lipolysis [3,4] but a decrease in catecholamine-stimulated lipolysis [5]. This study shows that depletion of WWP1 increases Adrb3 expression, which is decreased by an HFD, and increases phosphorylation of HSL. We speculate that WWP1, which is increased in obesity, is a candidate for reducing catecholamine-stimulated lipolysis due to obesity. This research will ultimately lead to a better understanding of a decrease in catecholamine-stimulated lipolysis in WAT in obesity. On the basis of our results, we anticipate that co-treatment of inhibition of WWP1 expression and Adrb3 agonists will become a novel treatment for catecholamine resistance of obesity and diabetes.

## Figures and Tables

**Figure 1 ijms-26-04219-f001:**
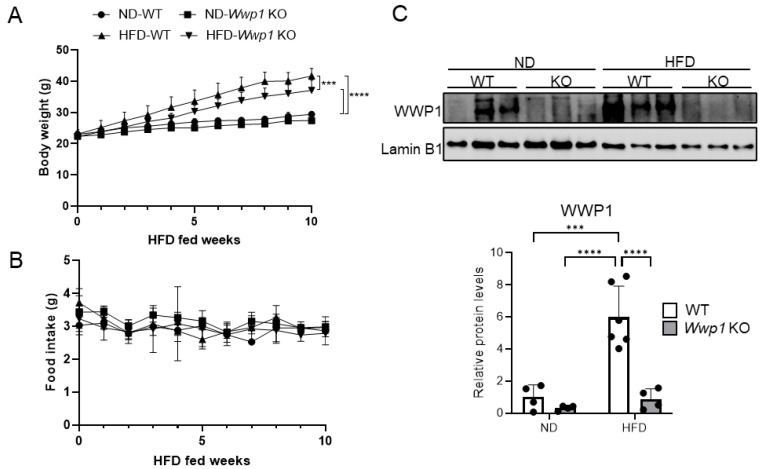
Body weight, food intake, and WWP1 protein levels of obese *Wwp1* KO mice. (**A**,**B**) Monitoring of body weight (*n* = 4–6) (**A**) and food intake (*n* = 10–14) (**B**) for 10 weeks in the HFD-fed period in 5-week-old ND- or HFD-fed Wwp1 KO mice. (**C**) WWP1 protein levels in eWAT derived from ND-WT, ND-*Wwp1* KO, HFD-WT, and HFD-*Wwp1* KO mice (upper panel; representative image, lower panel; quantitative value). Lamin B1 was used as an internal control. Each dot represents one mouse (*n* = 4–6). Quantitative values are the mean ± SD. Differences between these values were analyzed using two-way ANOVA and the Tukey–Kramer test (*** *p* < 0.001 and **** *p* < 0.0001).

**Figure 2 ijms-26-04219-f002:**
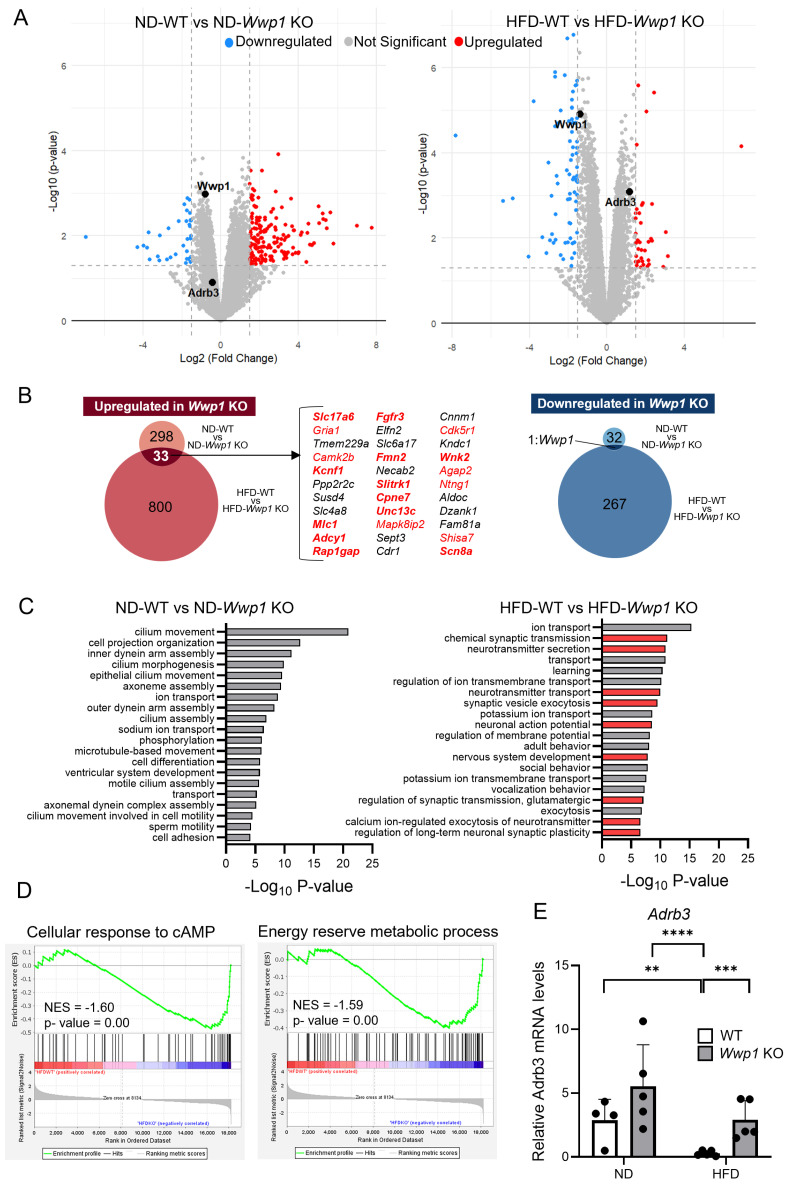
*Wwp1* KO increases nervous-system-related gene expression and *Adrb3* expression in eWAT of obese mice. (**A**–**D**) Expression analysis using RNA-seq of eWAT derived from ND-WT (*n* = 4), ND-*Wwp1* KO (*n* = 3), HFD-WT (*n* = 4), and HFD-*Wwp1* KO (*n* = 4) mice. Three or four samples in each group were used as experimental replicates. (**A**) Volcano plots present DEGs of eWAT of *Wwp1* KO mice compared with WT in the ND (left panel, total = 11,497 variables) and HFD (right panel, total = 10,779 variables) conditions. Red indicates upregulated genes (log2 fold change ≥ 1.5) and blue indicates downregulated genes (log2 fold change ≤ −1.5), with a *p* value threshold of 0.05. (**B**) Venn diagram showing overlapped upregulated (upper panel) and downregulated (lower panel) genes (false discovery rate < 0.1 and |FC| > 1.5) in WT and *Wwp1* KO mice in the ND and HFD conditions. The genes in red font are nervous-system-related factors. (**C**) Gene ontology enrichment analysis showing significantly enriched upregulated genes in *Wwp1* KO mice compared with WT mice in the ND and HFD conditions (false discovery rate < 0.1 and fold change > 1.5) using DAVID. Red bars indicate nervous-system-related pathways. (**D**) Gene set enrichment analysis of DEGs of eWAT of HFD-fed *Wwp1* KO mice compared with HFD-fed WT mice. (**E**) Levels of Adrb3 mRNA in WAT of ND- and HFD-fed WT and *Wwp1* KO mice. Rps18 was used as an internal control. Quantitative values are the mean ± SD. Each dot represents one mouse (*n* = 4–5). Differences between these values were analyzed using two-way ANOVA and the Tukey–Kramer test with log transformation (** *p* < 0.01, *** *p* < 0.001, and **** *p* < 0.0001).

**Figure 3 ijms-26-04219-f003:**
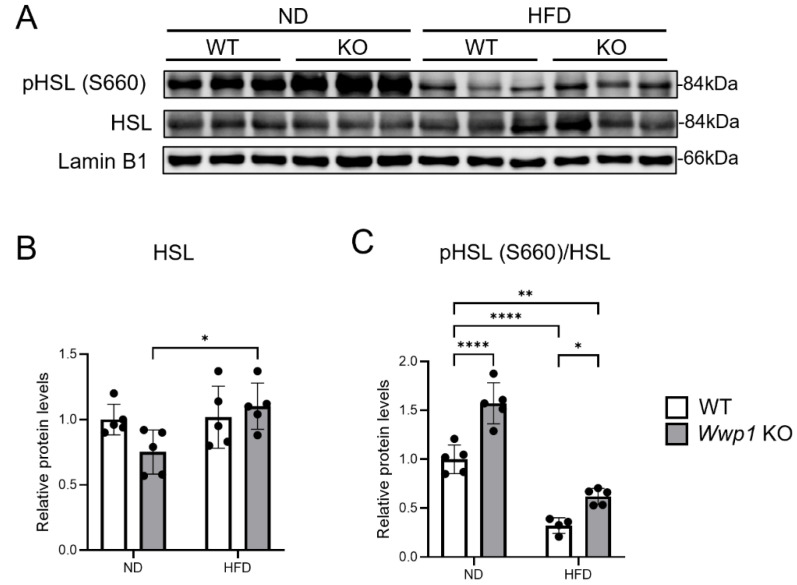
Depletion of WWP1 increases the expression levels of lipolysis-related proteins. (**A**) Protein expression levels of phospho-HSL (S660), HSL, and lamin B1 in eWAT of ND- and HFD-fed WT and *Wwp1* KO mice. Lamin B1 was used as an internal control. (**B**,**C**) Quantitative values of data shown in panel (**A**). The quantitative values are the mean ± SD. Each dot represents one mouse (*n* = 4–5). Differences between these values were analyzed using two-way ANOVA and the Tukey–Kramer test (* *p* < 0.05, ** *p* < 0.01, and **** *p* < 0.0001).

**Figure 4 ijms-26-04219-f004:**
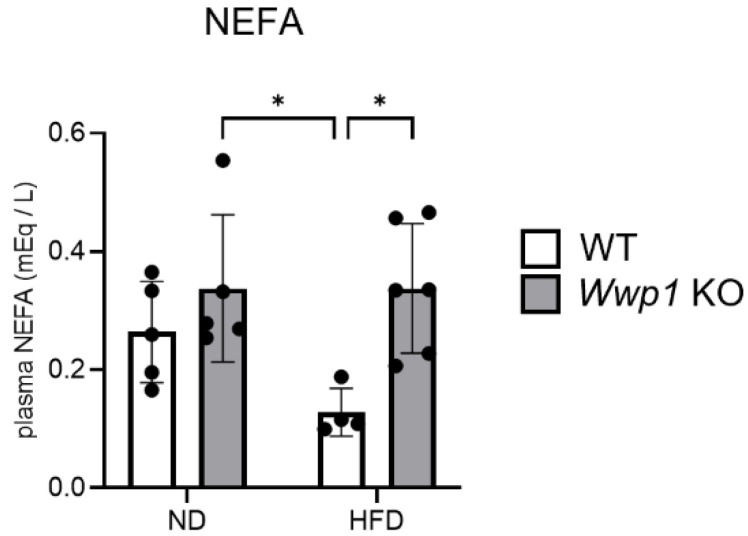
Depletion of WWP1 increases plasma NEFA concentrations. Plasma NEFA concentrations in ND- and HFD-fed WT and *Wwp1* KO mice are shown. The quantitative values are the mean ± SD. Each dot represents one mouse (*n* = 4–6). Differences between these values were analyzed using two-way ANOVA and the Tukey–Kramer test (* *p* < 0.05).

**Figure 5 ijms-26-04219-f005:**
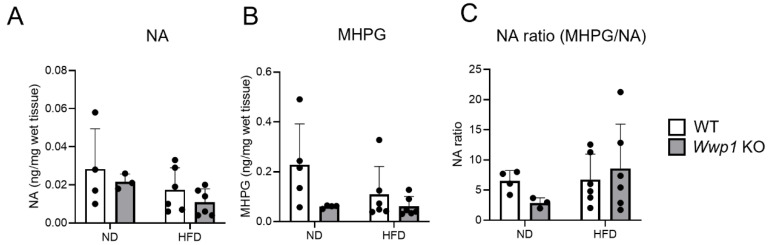
The MHPG/NA ratio is not changed by *Wwp1* KO in the HFD-fed condition. (**A**–**C**) NA (**A**), MHPG content (**B**), and the MHPG/NA ratio (**C**) in eWAT of ND- and HFD-fed WT and *Wwp1* KO mice. The quantitative values are the mean ± SD. Each dot represents one mouse (*n* = 4–6). Differences between these values were analyzed using two-way ANOVA and the Tukey–Kramer test.

**Table 1 ijms-26-04219-t001:** Organ weight of ND- or HFD-fed *Wwp1* KO mice.

	ND		HFD	
	WT	*Wwp1* KO	WT	*Wwp1* KO
Body Weight (g)	29.30 ± 0.97	27.38 ± 1.01	41.67 ± 2.45 ^####^	37.11 ± 3.23 * ^####^
sWAT (g)	0.37 ± 0.04	0.33 ± 0.16	2.04 ± 0.28 ^####^	2.32 ± 0.62 ^####^
eWAT (g)	0.62 ± 0.16	0.46 ± 0.14	1.64 ± 0.28 ^####^	1.92 ± 0.52 ^####^
Liver (g)	1.00 ± 0.03	1.02 ± 0.04	1.56 ± 0.32 ^###^	1.24 ± 0.17 *
QFM (g)	0.12 ± 0.03	0.13 ± 0.04	0.12 ± 0.04	0.13 ± 0.03
Heart (g)	0.12 ± 0.02	0.12 ± 0.01	0.13 ± 0.01	0.13 ± 0.01
Brain (g)	0.39 ± 0.04	0.39 ± 0.03	0.37 ± 0.03	0.39 ± 0.01

Values are the mean ± SD (*n* = 6–8). Differences between values were analyzed by two-way ANOVA and the Tukey–Kramer test. * *p* < 0.05 vs. WT, ^###^
*p* < 0.001, ^####^
*p* < 0.0001 vs. ND.

## Data Availability

The RNA-seq data have been deposited in the public repository “Gene Expression Omnibus” (GEO) (www.ncbi.nlm.nih.gov/geo (accessed on 17 December 2024)) and will be accessible through GEO: GSE286128. All data reported in this article will be shared by the lead contact upon request. Any additional information required to reanalyze the data reported in this article is available from the lead contact upon request.

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
