# Peer review of "Depletion of WWP1 Increases Adrb3 Expression and Lipolysis in White Adipose Tissue of Obese Mice"

_ijms, 2025, doi:10.3390/ijms26094219_

Round 1
Reviewer 1 Report
Comments and Suggestions for Authors
The article is devoted to a relevant topic of modern medicine. Authors used both one-way and two-way ANOVA tests, which should be mentioned in "Materials and methods" section. Secondly, ANOVA tests require normal (gaussian) distribution of variables in all study groups. Therefore, method of testing for normality and the statement that all compared groups had normal distribution sgould be added to "Materials and methods" section.
In "Results" section in Table 1 in line two ND should be replaced with WT.
The "reference" section should be reworked since most of the works cited by the artilce are rather outdated (>5 years old). With the exeption of references needed in "Materials and methods" section authors should updated (replace) the references fot newer ones.
Author Response
Responses to reviewer #1
We thank the reviewer for the helpful comments, which have been useful for improving our manuscript. Please find our responses to the comments and suggested revisions that we received below.
Authors used both one-way and two-way ANOVA tests, which should be mentioned in "Materials and methods" section.
As pointed out by the reviewer, we used a one-way ANOVA test only for data shown in Table 1. However, we should not have used a one-way ANOVA for these data. We believe that a two-way ANOVA would have been more appropriate for presentation of our data. Therefore, we re-analyzed these data using a two-way ANOVA. We have revised the Materials and Methods section and footnote of Table 1 accordingly.
Secondly, ANOVA tests require normal (gaussian) distribution of variables in all study groups. Therefore, method of testing for normality and the statement that all compared groups had normal distribution sgould be added to "Materials and methods" section.
On the basis of the reviewer’s comment, we performed the Shapiro–Wilk test to assess the normal (Gaussian) distribution in each group with and without log-transformation. However, because of the small sample size in our experiment, the test did not confirm normality. However, even with a small sample size, an ANOVA provides reliable results as long as the data are not highly skewed or have extreme outliers (Blanca et al., 2017). In addition to performing the Shapiro–Wilk test, we visually inspected Q-Q plots for each group. However, the Q-Q plots did not show any substantial departure from the normal distribution (see the attached figure). On the basis of this consideration and the fact that almost all data on lipolysis in our study were consistent, we decided to retain the ANOVA as the primary statistical test in our analysis.
To address the reviewer’s comment, we have revised the Materials and Methods section and stated that normality was tested but that an ANOVA was used on the basis of its robustness to non-normality as shown below.
Reference
Blanca, M. J., Alarcón, R., Arnau, J., Bono, R., & Bendayan, R. (2017). Non-normal data: Is ANOVA still a valid option? Psicothema, 29(4), 552-557. https://doi.org/10.7334/psicothema2016.383
In "Results" section in Table 1 in line two ND should be replaced with WT.
We thank the reviewer for pointing this out. We have revised Table 1 accordingly.
The "reference" section should be reworked since most of the works cited by the artilce are rather outdated (>5 years old). With the exeption of references needed in "Materials and methods" section authors should updated (replace) the references fot newer ones.
While we agree that citing recent literature is important, we have made a conscious effort to cite original articles that first reported the findings because we believe that acknowledging the primary sources of scientific discovery is important. Therefore, we have retained these original references wherever appropriate. We have referred to the original articles that formed the basis of a particular observation when possible.

Reviewer 2 Report
Comments and Suggestions for Authors
Obesity is associated with increased basal lipolysis but a decrease in catecholamine-stimulated lipolysis. The study conducted by the authors shows that depleting WWP1 leads to an increase in Adrb3 expression, which is reduced by a high-fat diet (HFD) and enhances the phosphorylation of hormone-sensitive lipase (HSL). The authors hypothesize that WWP1, which is elevated in cases of obesity, may play a role in reducing catecholamine-stimulated lipolysis in this condition. Their research aims to understand better the decreased catecholamine-stimulated lipolysis observed in obese white adipose tissue. Based on their findings, the authors anticipate that inhibiting WWP1 expression could be an effective strategy for treating obesity and diabetes alongside using Adrb3 agonists.
Strength of the manuscript: Establishing a new hypothesis based on experimental data. Adequate statistical analysis and representation of experimental data.
Weakness of the manuscript: The discussion is not followed by a scheme, so it is difficult to follow the text.
Suggested minor corrections:
- In the introductory part, explain in a few sentences how the accumulation of triglycerides leads to metabolic disorders.
- In parentheses or as an appendix, explain the meaning of WW domain from WW domain-containing E3 ubiquitin ligase 1.
- Transfer the sentence from the conclusion "Although Adrb3 agonists have been developed by several pharmaceutical companies as treatments for obesity and type-2 diabetes, but have yet to be successful" to the introduction and expand with the structural characteristics of the agonist, possibly showing several structures.
- The discussion would be much easier to follow if the receptors and interactions were shown schematically.
- In Figure 2 A, C and D part, it would be desirable to enlarge.
Author Response
Responses to reviewer #2
We thank the reviewer for the helpful comments, which have been useful for improving our manuscript. Please find our responses to the comments and suggested revisions that we received below.
Strength of the manuscript: Establishing a new hypothesis based on experimental data. Adequate statistical analysis and representation of experimental data.
Weakness of the manuscript: The discussion is not followed by a scheme, so it is difficult to follow the text.
We thank the reviewer for the helpful comment. We have revised the text in the Discussion section.
Suggested minor corrections:
- In the introductory part, explain in a few sentences how the accumulation of triglycerides leads to metabolic disorders.
WAT is mainly comprised of adipocytes, although other cell types contribute to its growth and function, such as pre-adipocytes, lymphocytes, macrophages, fibroblasts, and vascular cells. WAT in obesity is infiltrated by a large number of macrophages, and this recruitment is associated with systemic inflammation. Increased inflammation in WAT leads to the development of insulin resistance through chronic inflammation. In our RNA-seq data, the pathways related to immune responses were not enriched in Wwp1 KO (Fig. 2B). Changes in lipid metabolism in WAT of Wwp1 KO mice are thought to have little effect on chronic inflammation through metabolic disorders. To address the reviewer’s comment, we have revised the text by adding the above-mentioned information to the Introduction and Results sections.
- In parentheses or as an appendix, explain the meaning of WW domain from WW domain-containing E3 ubiquitin ligase 1.
WWP1 has a C2 domain in the N-terminal for membrane binding, WW1 domains in its central region for recognition of PPXY-containing substrate, and an HECT-domain in the C-terminal for the transfer of ubiquitin to substrate. We have revised the Introduction section accordingly.
- Transfer the sentence from the conclusion "Although Adrb3 agonists have been developed by several pharmaceutical companies as treatments for obesity and type-2 diabetes, but have yet to be successful" to the introduction and expand with the structural characteristics of the agonist, possibly showing several structures.
Mirabegron (Sacco & Bientinesi, 2012; Takasu et al., 2007) and vibegron (Di Salvo et al., 2017) were developed as Adrb3 agonists for treating overactive bladder. These Adrb3 agonists have been developed by several pharmaceutical companies as treatments for obesity and type-2 diabetes, but have yet to be successful. Nagiri and colleagues (Nagiri et al., 2021) published β3-adrenergic receptor (β3AR) structure complexes with mirabegron using cryogenic electron microscopy. They showed that these agonists had the basic pharmacophore 2-amino-1-phenylethanol, similar to adrenaline.
To address the reviewer’s comment, we have provided the reference for this study and revised the text accordingly in the Introduction section.
References
Sacco, E., & Bientinesi, R. (2012). Mirabegron: a review of recent data and its prospects in the management of overactive bladder. Therapeutic Advances in Urology, 4(6), 315-324. https://doi.org/10.1177/1756287212457114
Takasu, T., Ukai, M., Sato, S., Matsui, T., Nagase, I., Maruyama, T., Sasamata, M., Miyata, K., Uchida, H., & Yamaguchi, O. (2007). Effect of (R)-2-(2-aminothiazol-4-yl)-4'-{2-[(2-hydroxy-2-phenylethyl)amino]ethyl} acetanilide (YM178), a novel selective beta3-adrenoceptor agonist, on bladder function. J Pharmacol Exp Ther, 321(2), 642-647. https://doi.org/10.1124/jpet.106.115840
Di Salvo, J., Nagabukuro, H., Wickham, L. A., Abbadie, C., DeMartino, J. A., Fitzmaurice, A., Gichuru, L., Kulick, A., Donnelly, M. J., Jochnowitz, N., Hurley, A. L., Pereira, A., Sanfiz, A., Veronin, G., Villa, K., Woods, J., Zamlynny, B., Zycband, E., Salituro, G. M.,…Struthers, M. (2017). Pharmacological Characterization of a Novel Beta 3 Adrenergic Agonist, Vibegron: Evaluation of Antimuscarinic Receptor Selectivity for Combination Therapy for Overactive Bladder. J Pharmacol Exp Ther, 360(2), 346-355. https://doi.org/10.1124/jpet.116.237313
Nagiri, C., Kobayashi, K., Tomita, A., Kato, M., Kobayashi, K., Yamashita, K., Nishizawa, T., Inoue, A., Shihoya, W., & Nureki, O. (2021). Cryo-EM structure of the β3-adrenergic receptor reveals the molecular basis of subtype selectivity. Molecular Cell, 81(15), 3205-3215.e3205. https://doi.org/10.1016/j.molcel.2021.06.024
- The discussion would be much easier to follow if the receptors and interactions were shown schematically.
We have added a graphical abstract as Figure 6.
- In Figure 2 A, C and D part, it would be desirable to enlarge.
We thank the reviewer for the advice. We have revised Figure 2 as suggested.
Reviewer 3 Report
Comments and Suggestions for Authors
In this manuscript, Nozaki et al., assessed whether the depletion of WWP1 leads to the increase of Adrb3 expression and lipolysis in white adipose tissue.
The manuscript is well written and conclusions are supported by data.
I have a point of concern about the figure 2: why using different threshold/adjusted p-values in the volcano plot and in the venn diagram. It would be more understandable to use the same threshold for both representations (false discovery rate <0.1 and |FC|>1.5 for example).
In addition, what represents the “relative gene quantification” in PCR ? Which method was used to calculate this gene expression?
Author Response
Responses to reviewer #3
We thank the reviewer for the helpful comments, which have been useful for improving our manuscript. Please find our responses to the comments and suggested revisions that we received below.
I have a point of concern about the figure 2: why using different threshold/adjusted p-values in the volcano plot and in the venn diagram. It would be more understandable to use the same threshold for both representations (false discovery rate <0.1 and |FC|>1.5 for example).
We thank the reviewer for the helpful comment. We reanalyzed the Venn diagram using an FDR <0.1 and |FC| >1.5 and have revised Figure 2B.
In addition, what represents the “relative gene quantification” in PCR ? Which method was used to calculate this gene expression?
To calculate gene expression, we used the standard curve method. We prepared three-fold multiple pre-diluted cDNA samples for the standard curve and each cDNA sample. We then performed a PCR reaction using SYBR Green according to the manufacturer's protocols for Adrb3 and Rps18 as reference genes. A standard curve was then generated of the log concentration against Cq. The amount of cDNA in an unknown sample was calculated from its Cq value. Each Cq value of Adrb3 was compared with the amount of Rps18 to normalize the expression levels of the target gene across different samples. Finally, to determine how gene expression varies between samples, the values of the samples were plotted against the average value of ND-WT and presented as relative Adrb3 mRNA levels. In addition, we have changed the title of the Y-axis in Figure 2E from “Relative gene expression” to “Relative Adrb3 mRNA levels.”
Reviewer 4 Report
Comments and Suggestions for Authors
Manuscript ID: ijms-3470066
Title: Depletion of WWP1 increases Adrb3 expression and lipolysis in white adipose tissue of obese mice
In this study, the authors investigated the Adrb3 expression and lipolysis in Wwp1 systemic knockout mice. Overall, the study is so simple. And the experiments design is not scientific. First, the mouse experiments and protocols was not adequately described. The number of WT and KO mice (the number of replication, n) should be described clearly in the ND group or the HFD group. Generally, the number of replicates should be greater than 6 to achieve statistical significance in statistics. Furthermore, the growth curve of mice, food intake, body composition should present in results. And the conclusions can’t adequately support by the results. The expression of Adrb3 only detected by qPCR, should be further verified at the protein level. WWP1 as a new treatment of obesity and type-2 diabetes should verified by a series of experiments, such as ITT, GTT and so on. Finally, the Adrb3 activate of cAMP-PKA-HSL should verify by western blot. Therefore, I do not recommend to publish this article in this journal.
Author Response
Responses to reviewer #4
We thank the reviewer for the helpful comments, which have been useful for improving our manuscript. Please find our responses to the comments and suggested revisions that we received below.
First, the mouse experiments and protocols was not adequately described. The number of WT and KO mice (the number of replication, n) should be described clearly in the ND group or the HFD group. Generally, the number of replicates should be greater than 6 to achieve statistical significance in statistics.
We have added the number of WT and KO mice studied in each figure legend. The ANOVA has been shown to remain robust even when the assumption of normality is not strictly met, particularly in small-sample studies (Blanca et al., 2017). This previous study showed that mild deviations from normality did not significantly affect ANOVA results, making it a valid statistical approach, even for small datasets. Some of our data had a number less than 6, but according to Blanca et al., this is not a problem for statistical evaluation. Furthermore, almost all data in our study support the consensus that depletion of WWP1 increases lipolysis. Because of the robustness of the ANOVA, we decided to retain this test as the primary statistical method for group comparisons.
We have revised the Materials and Methods section by stating that normality was tested but that an ANOVA was used on the basis of its robustness to non-normality, particularly in small-sample studies.
Reference
Blanca, M. J., Alarcón, R., Arnau, J., Bono, R., & Bendayan, R. (2017). Non-normal data: Is ANOVA still a valid option? Psicothema, 29(4), 552-557. https://doi.org/10.7334/psicothema2016.383
Furthermore, the growth curve of mice, food intake, body composition should present in results.
In accordance with the reviewer’s comment, we have added the data of body weight during the HFD-fed period, added food intake to Supplementary Figure 1 (see attached figure), and revised the text of the Results section. Regarding body composition, we had already shown the individual tissue weights of ND-WT, ND-Wwp1 KO, HFD-ND, and HFD-Wwp1 KO mice at the time of euthanasia in Table 1 in the original manuscript.
the conclusions can’t adequately support by the results.
On the basis of the reviewer’s comment, we have added information regarding Adrb3 agonists to the Introduction section and revised our conclusion in the Discussion section.
The expression of Adrb3 only detected by qPCR, should be further verified at the protein level.
We agree that detecting protein levels of β3AR, which is coded by Adrb3, is important for a clear understanding. However, to the best of our knowledge, there are no antibodies that can detect β3AR in WAT. Therefore, we cannot measure protein levels of β3AR.
WWP1 as a new treatment of obesity and type-2 diabetes should verified by a series of experiments, such as ITT, GTT and so on.
In our previously study, ITT and GTT tests showed improved whole-body glucose metabolism in obese Wwp1 KO mice (Hoshino et al., 2020). Additionally, to assess the role of WWP1 in whole-body glucose metabolism, in another study we evaluated insulin signaling (pAkt/Akt rate) in insulin-sensitive tissues, such as WAT, the liver, and skeletal muscle. We found that obese Wwp1 KO mice showed an enhanced hepatic insulin signaling response and reduced weight and triglyceride contents only in the liver, but not in WAT (Nozaki et al., 2023). Therefore, targeting WWP1 may be useful in the treatment of obesity and type-2 diabetes through its dual functions. One of these functions is improving systemic glucose metabolism by increasing hepatic insulin sensitivity, and the other is increasing lipolysis in WAT by increasing Adrb3. We have added this information to the Discussion section.
References
Hoshino, S., Kobayashi, M., Tagawa, R., Konno, R., Abe, T., Furuya, K., Miura, K., Wakasawa, H., Okita, N., Sudo, Y., Mizunoe, Y., Nakagawa, Y., Nakamura, T., Kawabe, H., & Higami, Y. (2020). WWP1 knockout in mice exacerbates obesity-related phenotypes in white adipose tissue but improves whole-body glucose metabolism. FEBS Open Bio, 10(3), 306-315. https://doi.org/10.1002/2211-5463.12795
Nozaki, Y., Kobayashi, M., Wakasawa, H., Hoshino, S., Suwa, F., Ose, Y., Tagawa, R., & Higami, Y. (2023). Systemic depletion of WWP1 improves insulin sensitivity and lowers triglyceride content in the liver of obese mice. FEBS Open Bio, 13(6), 1086-1094. https://doi.org/10.1002/2211-5463.13610 (FEBS Open Bio)
Finally, the Adrb3 activate of cAMP-PKA-HSL should verify by western blot.
To evaluate the downstream signaling of Adrb3 in activating the cAMP-PKA-HSL pathway, we assessed PKA activation by detecting phosphorylated PKA substrates with a phospho-PKA substrate antibody in eWAT of obese Wwp1 KO mice. However, phosphorylated PKA substrates levels were not significantly changed (see attached figure). cAMP signaling is not uniformly distributed within the cell but instead occurs in spatially confined microdomains. In compartmentalization of cAMP signaling, largely mediated by A-kinase anchoring proteins, the compartmentalization plays a critical role in determining the specificity and functional outcome of PKA activation (Steinberg & Brunton, 2001; Zaccolo, 2011). Therefore, compartmentalization and local control of cAMP may be necessary for lipolysis signaling downstream of Adrb3 by WWP1. In accordance with the reviewer’s comment, we have added western blot data using phospho-PKA substrate (Supplementary Figure 2) and revised the text in the Discussion section.
References
Steinberg, S. F., & Brunton, L. L. (2001). Compartmentation of G protein-coupled signaling pathways in cardiac myocytes. Annu Rev Pharmacol Toxicol, 41, 751-773. https://doi.org/10.1146/annurev.pharmtox.41.1.751
Zaccolo, M. (2011). Spatial control of cAMP signalling in health and disease. Current Opinion in Pharmacology, 11(6), 649-655. https://doi.org/10.1016/j.coph.2011.09.014

Round 2
Reviewer 4 Report
Comments and Suggestions for Authors
Manuscript ID: ijms-3470066
Title: Depletion of WWP1 increases Adrb3 expression and lipolysis in white adipose tissue of obese mice
The author's response has partially addressed my concerns, but the key issues have not been effectively resolved. First,I disagree the opinion:Some of our data had a number less than 6, but according to Blanca et al., this is not a problem for statistical evaluation. Furthermore, I find it unclear why the authors relegated the critical results on body weight and feed intake to the supplementary materials. The authors' characterization of body composition appears problematic. These measurements (e.g., adipose tissue mass and lean body mass) should have been determined through quantitative nuclear magnetic resonance (NMR) analysis in mice, as per established methodologies. The fundamental limitation lies in the absence of conclusive evidence proving that Adrb3 drives lipolysis through cAMP-PKA-HSL signaling — a core mechanistic claim central to this study.
Author Response
We thank the reviewer for the helpful comments again, which have been useful for improving our manuscript. Please find our responses to the comments and suggested revisions that we received below. In addition, revisions made to the text based on the comments received are indicated in blue in main manuscript.
The author's response has partially addressed my concerns, but the key issues have not been effectively resolved. First,I disagree the opinion:Some of our data had a number less than 6, but according to Blanca et al., this is not a problem for statistical evaluation.
We appreciate the reviewer’s comment regarding the assumption of normality. In our previous response, we referenced Blanca et al. (2017) to support the use of ANOVA in small-sample studies. However, we acknowledge that our phrasing may have overstated the conclusions of that study. Blanca et al. suggest that ANOVA can be robust to deviations from normality, even when sample sizes are small, but they do not explicitly state that it is appropriate in all cases when n < 6.
To clarify and reinforce this point more appropriately, we have now included a reference to Norman (2010), who emphasized that “parametric methods are remarkably robust in most practical situations,” even when assumptions such as normality are not strictly met.
As reviewer’s correctly pointed out, due to the small sample size in our experiment, the Shapiro-Wilk test did not provide statistically significant evidence of normality. However, we also inspected the Q-Q plot for each group, and the plots did not show any substantial departure from the normal distribution. Based on this visual confirmation, we believe that assuming approximate normality is reasonable.
Based on the above paper’s opinions and our Q-Q plots data, we believe that the use of ANOVA remains statistically justified in this context. To address the reviewer’s comment, we have further revised the Statistical Analysis section and added the Q-Q plots as Supplementary data (Supplementary Figure 3).
Reference:
Norman, G. (2010). Likert scales, levels of measurement and the "laws" of statistics. Adv Health Sci Educ Theory Pract, 15(5), 625-632. https://doi.org/10.1007/s10459-010-9222-y
Furthermore, I find it unclear why the authors relegated the critical results on body weight and feed intake to the supplementary materials. The authors' characterization of body composition appears problematic. These measurements (e.g., adipose tissue mass and lean body mass) should have been determined through quantitative nuclear magnetic resonance (NMR) analysis in mice, as per established methodologies.
Regarding the use of NMR analysis in mice for measurement body composition, we agree that this is a standard method. However, obtaining NMR data of 10 weeks period high-fat fed mice of Wwp1 KO mice would require generating a new cohort specifically for only NMR imaging purposes. Due to concerns regarding the ethical use of animals in research, we have considered it unjustifiable to create additional experimental animals solely for this purpose, particularly given that detailed body composition analysis using NMR is not central focus of this study.
Nevertheless, we acknowledge the validity of your concern, we have moved the data on whole body weight and food intake to the main figure (Figure 1A and B, respectively). We measured the weights of subcutaneous and gonadal adipose tissue depots; however, the retroperitoneal fat pad was not collected. As a result, we were unfortunately unable to accurately calculate total fat mass or lean body mass in this study.
The fundamental limitation lies in the absence of conclusive evidence proving that Adrb3 drives lipolysis through cAMP-PKA-HSL signaling — a core mechanistic claim central to this study.
As the reviewer rightly pointed out, our initial interpretation regarding the activation of lipolysis via the cAMP–PKA–HSL pathway may have been overstated, because the levels of phosphorylated PKA substrates were not significantly altered in obese Wwp1 KO mice. In response to the reviewer’s comment, we have revised the relevant text to reflect a more cautious interpretation of the signaling pathway in the abstract section.
Importantly, the central claim of our manuscript is that depletion of WWP1 restores Adrb3 expression, which is otherwise suppressed by obesity, and enhances HSL phosphorylation. Furthermore, the increased plasma NEFA concentrations observed in HFD-fed Wwp1 KO mice (Fig. 4) support the conclusion that lipolysis is enhanced in these animals.
Regarding the signaling mechanism, we acknowledge that our data do not provide direct evidence that the cAMP–PKA–HSL pathway mediates this effect. However, we have addressed this limitation in the Discussion section. Specifically, we note that cAMP/PKA signaling is highly compartmentalized within cells and may not be detectable through whole-tissue Western blot analysis. This mechanistic consideration, and the possibility of spatially restricted PKA activity downstream of Adrb3, is already discussed in detail in the revised manuscript (see Discussion, paragraph 2).
Round 3
Reviewer 4 Report
Comments and Suggestions for Authors
The author's response has partially addressed my concerns, but the key issues have not been effectively resolved. First, I keep my previous comments. Generally, the number of replicates should be greater than 6 to achieve statistical significance in statistics. Furthermore, what confuses me is that in the text, the number of sample replicates alternates between 6-8, 4-6, and 10-14. I remain particularly curious about the effects of Wwp1 KO on murine skeletal muscle. Generally, investigations into adipose tissue should include assessments of lean mass impacts. Furthermore, if the proposed mechanism involves HSL upregulation promoting lipolysis, it would be essential to conduct energy metabolism profiling via metabolic cages and evaluate thermogenic activity using infrared thermography.